# Safety and effectiveness of hormonal vs non-hormonal or no contraception in women with hypertension and future fertility desire: A broad-scope systematic review

Natalia Losada-Trujillo[1,2,3,4*☯], Kelly Estrada-Orozco[1,2,4,5☯],
Oscar Julián Velasco-Lancheros[1,3,4☯], Brahiam Alejandro Ramírez-Vargas[1,3,4☯],
Álvaro Javier Burgos-Cárdenas[1,2,3,6], Paula Andrea González-Caicedo[1,2,3‡],
María José Hoyos Bedoya[1,2,3‡], Hernando Gaitán-Duarte[1,2,3,4☯]

1 Faculty of Medicine, Universidad Nacional de Colombia, Bogotá D.C, Colombia, 2 Clinical Research Institute, Faculty of Medicine, Universidad Nacional de Colombia, Bogotá D.C, Colombia, 3 Hospital Universitario Nacional de Colombia, Bogotá D.C, Colombia, 4 Health Technologies and Policies Assessment Group (GETS), Bogotá D.C, Colombia, 5 Centro de Evidencia e Implementación, CEI-evidence, Bogotá D.C, Colombia, 6 Fundación Cardioinfantil, Instituto de Cardiología, Bogotá D.C, Colombia

☯ These authors contributed equally to this work
‡ AJBC, PAGC, MJHB also contributed equally to this work
* nlosadat@unal.edu.co

## Abstract

### Objective

To evaluate the safety and effectiveness of hormonal contraceptives compared to non-hormonal methods or no contraception in hypertensive women seeking future fertility.

### Data sources

Searches were conducted in Medline, Embase, CENTRAL, and LILACS (September–October, 2022), with updates on September, 2023, and August, 2024. Clinical trial records, regulatory agencies, and gray literature were also consulted (June, 2022; August, 2023; September, 2024).

### Methods

Eligible studies that assessed safety (major adverse cardiovascular events, pelvic inflammatory disease, vaginal infections, loss of fertility, discontinuation, worsening hypertension, peripheral arterial disease, venous thromboembolism, weight gain, liver or kidney function changes) or effectiveness (pregnancies/Pearl index) of hormonal contraceptives in hypertensive women. Eligible designs included clinical trials, cohort and case control studies, case series, case reports, and adverse event data. Two reviewers independently screened titles/abstracts, reviewed full

**Data availability statement:** All data underlying the findings of this systematic review are fully available without restriction. The files containing the articles screened by title and abstract from the databases, provided in standard reference formats (.ris, .bib, .csv, and .enw), as well as the presentation summarizing the study, are available in the Zenodo repository (https://doi.org/10.5281/zenodo.18906040; https://doi.org/10.5281/zenodo.18906289). Additional materials supporting the study, including search strategies, data extraction forms, quality assessment results, and supplementary tables, are provided as Supporting information files accompanying this paper.

**Funding:** The author(s) received no specific funding for this work.

**Competing interests:** The authors have declared that no competing interests exist.

texts, selected studies, extracted data, and assessed bias (ROBINS-I: cohorts), methodological quality (Newcastle Ottawa: case-control), and critical appraisal (Joanna Briggs Institute: case series). Certainty of evidence was evaluated using GRADE. Evidence was synthesized quantitatively and qualitatively by contraceptive and outcome.

## Results

Of 32,225 records screened, 17 studies were included. Combined oral contraceptives may increase hemorrhagic cerebrovascular disease (OR: 1.64 (95% CI 1.08–2.50, certainty: low). Evidence is very uncertain about their effects on preventing pregnancies, and safety related to ischemic cerebrovascular disease, myocardial infarction, venous thromboembolism, weight gain, hypertension, kidney function, and lipid profile. Similarly, the effects of progestin-only pills, injectables, and vaginal rings on cardiovascular and metabolic outcomes remain highly uncertain. Certainty is also very low for combined oral contraceptives or progestogen-only pills regarding cerebrovascular disease, myocardial infarction (OR: 1.15, 95% CI 0.60–2.19), and peripheral arterial disease.

## Conclusions

Low-certainty evidence suggests that combined oral contraceptives may increase hemorrhagic cerebrovascular disease in hypertensive women, whereas evidence for the other assessed safety and effectiveness outcomes is very low.

---

## 1. Introduction

Global life expectancy has increased as well as chronic non-communicable diseases [1] and the age of women at childbirth [2]. In 2021, the Organization for Economic Cooperation and Development (OECD) reported an average childbirth age of 30.9 years in its member countries [3]. Higher age raises the risk of chronic diseases [1], and pregnancy-related complications [4]. In 2019, in women of reproductive age (15–49 years), the main causes of death from non-communicable diseases worldwide were: neoplasms (19.66%), cardiovascular diseases (15.03%), digestive diseases (5.39%) and diabetes/chronic kidney disease (4.53%) [1]. Hypertension is the main cardiovascular risk factor in this population [1], which is associated with Major Adverse Cardiovascular Events (MACE) (cardiovascular death, acute myocardial infarction (AMI), cerebrovascular disease (CVD), unstable angina, coronary revascularization, acute heart failure or worsening of heart failure, and transient ischemic attack (TIA) [5,6]). In 2019, ischemic heart disease caused 6.31% of non-communicable disease deaths, while ischemic CVD accounted for 0.75% of deaths in women of reproductive age [1].

Studies suggest hypertensive women using hormonal contraceptives have higher cardiovascular risk. Estrogens may enhance blood coagulability by raising

coagulation factors II, VII, VIII, IX, and X, fibrinogen, and soluble fibrin, while reducing antithrombin III and vascular wall fibrinolytic activator [7,8]. Hypertensive women need safe and effective contraceptive methods. Hormonal methods are very effective but may increase cardiovascular risk in high-risk women [8]. Evidence on the safety and effectiveness of all hormonal contraceptives is limited. Two critically low-quality systematic reviews [9,10] have evaluated the safety of only combined oral contraceptives (COCs) in hypertensive women. Current World Health Organization (WHO) recommendations [8] rely on individual studies and expert consensus rather than systematic reviews. This lack of high-quality evidence highlights the need for a systematic review of the safety and effectiveness of hormonal contraceptives in hypertensive women to guide clinical decisions. The characteristics of hormonal and non-hormonal methods are presented in S1–S2 Appendix.

This systematic review objective is to evaluate the safety and effectiveness of hormonal versus non-hormonal or non-use of contraception in women of reproductive age with a desire for future fertility and hypertension.

## 2. Methods

The protocol of this systematic review was registered in the International Prospective Register of Systematic Reviews (PROSPERO CRD42022324806). Results were reported according to the Preferred Reporting Items for Systematic Reviews and Meta-analyses (PRISMA) and Meta Analysis of Observational Studies in Epidemiology (MOOSE) reporting guidelines [11–13] (S3–S5 Appendices). Likewise, this systematic review was carried out with the guidelines of the Cochrane Handbook for Systematic Reviews of Interventions [14].

### a. Eligibility criteria

There were no language or publication date restrictions. The intervention group comprised hormonal contraceptive users, while the comparator included non-hormonal or non-use of contraceptives.

**Inclusion criteria:** Studies assessing the safety (clinical trials (RCTs), cohorts, case-controls, case reports, case series, clinical trial records, and adverse event reports) or effectiveness (RCTs and cohorts) of hormonal contraception in hypertensive women of reproductive age (15–49 years) desiring future fertility.

**Exclusion criteria:** Studies exclusively comparing only hormonal contraceptives with each other, and those including women with polycystic ovary syndrome, abnormal uterine bleeding, heavy menstrual bleeding, or early ovarian failure (hormone replacement).

Reviews, meta-analyses, guidelines, expert consensus, abstracts, editorials, and comments were not considered for inclusion. Definitive surgical methods were not considered, as the focus was on women preserving fertility.

Outcomes considered were: Primary: MACE (death, AMI, CVD, hospitalization for unstable angina, need for coronary revascularization, hospitalization for acute heart failure or worsening of existing heart failure and TIA [5,6,15]) and unwanted pregnancies. Secondary: Pearl Index, pelvic inflammatory disease (PID), vaginal infections, infertility, contraception discontinuation due to side effects or drug interactions, worsening hypertension, peripheral arterial disease, venous thromboembolism (VTE), weight gain, and alteration in liver or kidney function.

### b. Information sources and search strategies

The initial search (September 29–October 4, 2022) was updated on September 12–13, 2023, and August 7–8, 2024, in the databases Medline (via Ovid), Embase, Cochrane Central Registry of Controlled Trials (CENTRAL) and the database of Latin American and Caribbean Literature in Life Sciences Health (LILACS). Additionally, on June 27, 2022, August 26–27, 2023, and September 11–12, 2024, searches were carried out in the registries of clinical trials, regulatory agencies, and databases specialized in reporting adverse events, post-marketing safety and gray literature bases. A manual

search of the references in the included studies was conducted to ensure all relevant studies were identified. The sources of information are in S6 Appendix, and the search strategies with updates are in S7 Appendix.

### c. Study selection

After searching the databases, the records were uploaded into Rayyan [16] to remove duplicates. Then, two reviewers independently screened titles and abstracts, following the eligibility criteria. The selected studies were reviewed in full text by two reviewers independently. In case of discrepancies, a third reviewer was consulted. The reasons for not including studies that were read in full text are found in S8 Appendix. This process was also applied to gray literature, clinical experiment records, and safety reports on their respective platforms.

### d. Data extraction and management

Information from the included full-text studies was extracted independently by two reviewers (NL, AB, MH, PG) on the REDCap platform [17]. Data extraction was performed independently. When differences occurred in the extracted data, they were resolved through dialogue and reviewing the evaluated study jointly. One reviewer transferred the collected data to Review Manager 5.4.1 [18]. In those cases in which specific data from the studies were not available, we sought to contact the authors of those studies. The items on the data extraction forms for each study design are found in S9 Appendix.

### e. Evaluation of the risk of bias, methodological quality and critical approach of the included studies

Two reviewers (NL, AB, MH, PG) independently assessed the risk of bias in the cohort studies (ROBINS-I) [19], the methodological quality of the case-control studies (Newcastle-Ottawa) [20], and the critical approach of the included case series studies (Joanna Briggs Institute checklist) [21]. This information was compiled in REDCap [17]. Any disagreement was resolved through dialogue.

The certainty of evidence of each outcome was evaluated using the GRADE approach according to the type of study design [22].

### f. Measures of the effect

For the dichotomous primary and secondary outcomes, we sought to use estimators adjusted for possible confounding variables that would present the relationship between the exposure and the outcome in hypertensive women. In the absence of these, Odds Ratios (OR) were calculated with their respective confidence intervals as measures of association. OR were estimated instead of RR since it is a more stable measure of association, and since the outcomes evaluated are rare in the population of interest, the OR does not overestimate the association. It was not possible to calculate OR adjusted for potential confounding variables, as we did not have the original data from the studies.

If the study did not report or a measure of association could not be calculated, narrative synthesis was performed. For continuous outcomes, narrative synthesis was performed because it was not possible to obtain from the studies the standard deviations or standard errors of the difference in means between the two groups that compared the studies, nor p value, t statistic, confidence intervals of the differences in intergroup averages.

### g. Handling of unavailable data

In the presence of missing data, we tried to contact the authors of the studies to recover the information, however the response was not available.

### h. Subgroup analysis and heterogeneity assessment

Subgroup analysis was conducted by type of hormonal contraceptive and by "current use" definition, considering prior use duration at the outcome index date in cases of considerable heterogeneity. Analysis by hypertension severity was planned but not performed due to lack of data.

Clinical, methodological, and statistical heterogeneity were assessed. Statistical heterogeneity was identified by evaluating confidence interval overlap and Chi² test p-values (<0.05, or <0.1 for small or few studies). Following the Cochrane Handbook, I² values were interpreted as: 0–29%: not important, 30–49%: moderate, 50–74%: substantial, and 75–100%: considerable heterogeneity [23].

### i. Publication bias assessment

It was considered that if there were 10 or more studies, it would be evaluated using graphical methods (funnel plot) or statistical methods, when the number was less than 10 studies, we use the Tool for assessing Risk Of Bias due to Missing Evidence in a meta-analysis (ROB-ME) [24].

### j. Data synthesis

Evidence was synthesized by hormonal contraceptive type and outcome, using quantitative methods (association estimators, vote counting, meta-analysis) and qualitative methods (narrative synthesis) when meta-analysis was not possible. Outcomes were presented by contraceptive type and study design.

Meta-analysis was conducted using Review Manager 5.4.1 [18], when at least two studies per outcome were available, with complete dichotomous outcome data, consistent effect measures, similar participants, exposure, and research questions, and no considerable heterogeneity. The approach considered clinical, methodological, and statistical diversity. When feasible, a random-effects meta-analysis using the DerSimonian and Laird method was performed to enhance generalizability, account for multiple exposure effects, and incorporate study heterogeneity [25].

For dichotomous outcomes in which meta-analysis was not performed, crude OR with their 95% confidence intervals (95%CI) were calculated using Stata 15 statistical software [26], and the OpenEpi software [27], if there was the information necessary for the calculation. For all outcomes in which meta-analysis could not be performed, synthesis was performed by vote counting based on the direction of effect [25]. Each effect estimate was classified as favoring the outcome, against the outcome, or showing no difference. Vote counting consisted of tallying the number of studies contributing to each direction of effect for a given outcome and was interpreted in conjunction with the certainty of the evidence. The continuous outcomes were synthesized qualitatively.

Summary tables were created using the GRADE approach to assess the certainty of evidence. Two reviewers (NL, AB) independently evaluated outcomes based on GRADE domains using GRADEpro GDT software [28]. The data synthesis structure is shown in the diagram in S10 Appendix.

### k. Sensitivity analysis

For the sensitivity analysis, to evaluate the robustness of the conclusions, outcomes were compared by study design, considering risk of bias, methodological quality, or critical approach as appropriate.

### l. Modifications to the protocol

• We added the category "current use" vs. "non-current use" of hormonal contraceptives as a new comparison based on data from case-control studies.

• Secondary outcomes were added: peripheral arterial disease and alteration of metabolic parameters.

• A new exposure category called "combined and progestin-only hormonal contraceptives" was created.

## 3. Results

### a. Description of studies

A total of 32,225 records were identified from databases, clinical experiment databases, regulatory agencies, and gray literature (sources in S6 Appendix). From the databases (Medline, Embase, CENTRAL, LILACS), 26,454 records were identified. After removing 7,017 duplicates, 19,437 records were screened by title and abstract, selecting 65 for full-text review. Of these, 47 were not included [29–75] (reasons in S8 Appendix) getting 17 studies for analysis: 13 case-control [76–89], 2 were cohort [90,91] and 2 case series [92,93]. We did not find RCTs.

A total of 5,771 references were identified from clinical experiment databases, regulatory agencies, and gray literature, but none were included (Fig 1).

### b. Studies characteristics

13 case-control studies were included [76–89], 2 cohort studies [90,91] and 2 case series [92,93]. The design of the studies, the types of hormonal contraceptives and their comparators, as well as the outcomes evaluated, are detailed in Table 1. The detailed characteristics of the studies included are in S11 Appendix, definitions of exposures and interventions are in S12 Appendix.

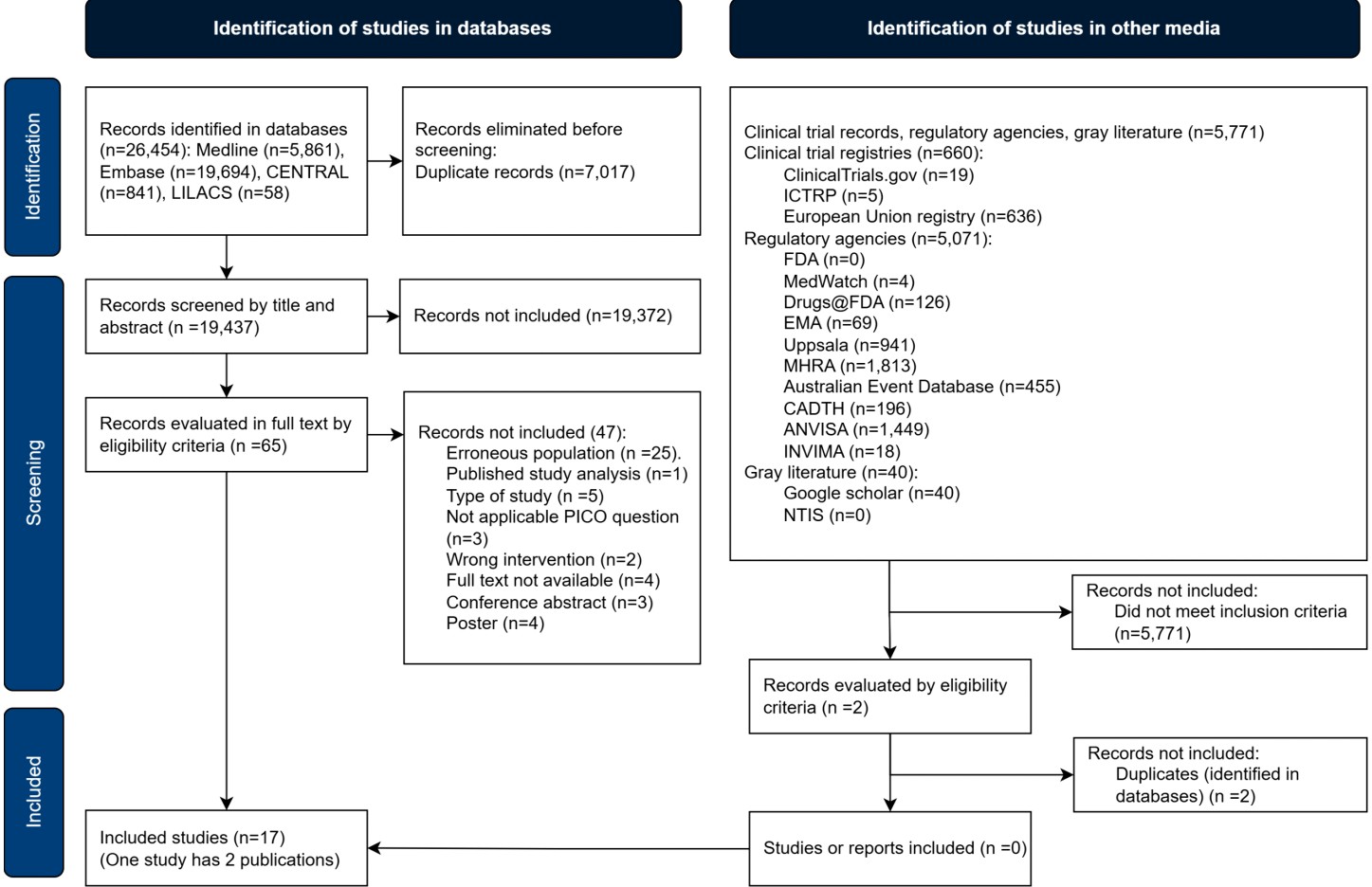

**Fig 1. PRISMA flowchart.** The study selection process and information sources are detailed in the PRISMA flowchart.

**Table 1. General characteristics of the included studies.**

| Study | Design | Population | | | Intervention/ Exposure | Comparator | Outcomes assessed |
|---|---|---|---|---|---|---|---|
| | | n | Age (years)[1] | % with hyper-tension[2] | | | |
| Collaborative Group 1975 [84] | Case-control study. | 399 | Overall range: 15–44 | 32.81 | Current use of combined oral contraceptives | No current use of com-bined oral contraceptives | Ischemic or hemor-rhagic cerebrovas-cular event |
| Lidegaard 1993 and 1995 [86,87] | Case-control study. | 124 | Overall range: 15–44 | 66.42 | Current use of com-bined or progestin-only oral contraceptives | No current use of com-bined or progestin-only oral contraceptives | Ischemic cerebro-vascular event |
| Croft 1989 [77] | Case-control study nested in a cohort. | 93 | Cases: Range: 20–59 (96.2%), ≥60 (3.8%). Controls: Range: No data | 14.72 | Current use of com-bined or progestin-only oral contraceptives | No current use of com-bined or progestin-only oral contraceptives | Acute myocardial infarction |
| Hannaford 1994 [78] | Case-control study nested in a cohort. | 134 | Overall range: 21–70 Cases: Median: 45.4 Controls: No data | 13.24 | Current use of com-bined or progestin-only oral contraceptives | No current use of com-bined or progestin-only oral contraceptives | Ischemic and hem-orrhagic cerebrovas-cular event |
| Tanis 2001 [83] | Case-control study. | 115 | Overall range: 18–49 Cases: Mean (SD): 42.7(6.5) Controls: Mean (SD): 38.1(8.3) | 9.8 | Current use of com-bined or progestin-only oral contraceptives | No current use of com-bined or progestin-only oral contraceptives | Acute myocardial infarction |
| Kemmeren 2002 [89] | Case-control study. | 104 | Overall range: 18–49 years. Cases: Mean (SD): 39.3(8.3) Controls: Mean (SD): 38.1(8.3) years | 9.22 | Current use of com-bined or progestin-only oral contraceptives | No current use of com-bined or progestin-only oral contraceptives | Ischemic cerebro-vascular event |
| Van Den Bosch 2003 [81] | Case-control study. | 101 | Overall range: 18–49 years. Cases: Mean (SD): 43.7(5.8) Controls: Mean (SD): 38.1(8.3) | 1019.4 | Current use of com-bined or progestin-only oral contraceptives | No current use of com-bined or progestin-only oral contraceptives | Peripheral arterial disease |
| Heinemann 1998 [76] | Case-control study. | 85 | Cases: Range, n (%): 16–24: 31 (14.1), 25–34: 77 (35.0), 35–44: 112 (50.9) Hospital controls: Range, n (%): 16–24: 46 (13.7), 25–34: 117 (34.8), 35–44: 173 (51.5) Community controls: Range, n (%): 16–24: 61 (13.9), 25–34: 158 (36.0), 35–44: 220 (50.1). | 8.54 | Current use of combined oral contraceptives | No current use of com-bined oral contraceptives | Ischemic cerebro-vascular event |

*(Continued)*

**Table 1.** (Continued)

| Study | Design | Population | | | Intervention/ Exposure | Comparator | Outcomes assessed |
|-------|--------|-----------|---|---|------------------------|------------|-------------------|
| | | n | Age (years)[1] | % with hyper-tension[2] | | | |
| WHO 1995 [79] | Case-control study. | 201 | Cases: Europe: mean (SD): 32.5 (7.0). Developing countries: mean (SD): 32.7 (7.3). Controls: Europe: mean (SD): 32.2 (7.0). Developing countries: mean (SD): 32.4 (7.3). | 4.85 | Current use of combined oral contraceptives | No current use of com-bined oral contraceptives | Venous thromboembolism |
| WHO 1996a [85] | Case-control study. | 495 | Cases: Europe: mean (SD): 36.3 (6.0). Developing countries: mean (SD): 36 (6.5). Controls: Europe: mean (SD): 35.9 (6.1). Developing countries: mean (SD): 35.6 (6.6). | 12.44 | Current use of combined oral contraceptives | No current use of com-bined oral contraceptives | Hemorrhagic cere-brovascular event |
| WHO 1996b [88] | Case-control study. | 276 | Cases: Europe: mean (SD): 35.8 (5.8). Developing countries: mean (SD): 35.3 (6.5). Controls: Europe: mean (SD): 35.5 (5.9). Developing countries: mean (SD): 34.9 (6.5). | 10.44 | Current use of combined oral contraceptives | No current use of com-bined oral contraceptives | Ischemic cerebro-vascular event |
| WHO 1997 [82] | Case-control study. | 148 | Cases: Range, n (%): Europe: < 35 (18.7), 35–39 (26.8), ≥ 40: 54.5). Developing coun-tries: < 35: (20.6), 35–39: (28.8), ≥ 40 (50.6). Controls: Range, n (%): Europe: < 35 (21.3), 35–39 (27.5), ≥ 40 (51.2). Developing coun-tries: < 35: (21.7), 35–39(30.2), ≥ 40 (48.1). | 11.31 | Current use of combined oral contraceptives | No current use of com-bined oral contraceptives | Acute myocardial infarction |

*(Continued)*

**Table 1.** (Continued)

| Study | Design | Population | | | Intervention/ Exposure | Comparator | Outcomes assessed |
|---|---|---|---|---|---|---|---|
| | | n | Age (years)[1] | % with hyper- tension[2] | | | |
| WHO 1998 [80] | Case-control study | 1325 | Cases: No data<br><br>Controls: Exposed to progestin-only oral contraceptives: mean (SD): 31.8 (7.1). Exposed to combined injectables: mean (SD): 32.6 (5.5). Exposed to progestin-only injectables: mean (SD): 31 (6.2). Unexposed: mean (SD): 35.4 (6.8). | 9.68 | Current use of progestin-only oral contraceptives, progestin-only injectables, or combined injectable contraceptives | No current contraceptive use of progestin-only oral contraceptives, progestin-only injectables, or combined injectable contraceptives | Ischemic and hemorrhagic cerebrovascular events, acute myocardial infarction and venous thromboembolism |
| de Morais 2014 [90] | Cohort study. | 56 | No data. | 100 | Use of combined oral contraceptives | Use of condom or copper intrauterine device | Unwanted pregnancies, worsening of the underlying condition, increase in body mass, alteration of kidney function tests and deterioration of metabolic parameters |
| de Rossi 2014 [91] | Cohort study. | 65 | No data | 100 | Use of combined oral contraceptives | Use of condom or copper intrauterine device | Worsening the baseline condition, increase in body mass and deterioration of metabolic parameters |
| Elkik 1986 [93] | Case series study. | 12 | Mean (SD): 29.17 (6.84). | 100 | Use of combined vaginal ring | NA[3] | Worsening of the underlying condition |
| Bounhoure 2008 [92] | Case series study. | 3 | The 3 hypertensive women were: 30, 34 and 36 years old. | 25 | Use of combined oral contraceptives | NA[3] | Acute myocardial infarction |

[1] Presented as range, mean (standard deviation (SD)), or median (interquartile range (IQR)), depending on the information available in the study; [2]Percentage of hypertensive women in the study; [3] NA: do not apply to the study.

### c. Assessment of the risk of bias, methodological quality and critical approach of the included studies

The 13 case-control studies [76–89] presented good methodological quality, the 2 cohort studies [90,91] presented a critical risk of bias and the 2 case series [92,93] presented low critical quality. In Tables 2–4 the evaluation of methodological quality, risk of bias and critical appraisal of these studies are presented. The details of these evaluations are presented in S11 Appendix. The evaluation of the certainty of the evidence of the outcomes for individual and grouped studies, using the GRADE approach, is found in S13–S14 Appendices.

**Table 2. Evaluation of the methodological quality of case-control studies.**

| Study | Collaborative Group 1975 [84] | Lidegaard 1993,1995 [86,87] | Croft 1989 [77] | Hannaford 1994 [78] | Kemmeren 2002 [89] | Tanis 2001 [83] | Van Den Bosch 2003 [81] | Heinemann 1998 [76] | WHO 1995 [79] | WHO 1996a [85] | WHO 1996b [88] | WHO 1997 [82] | WHO 1998 [80] |
|---|---|---|---|---|---|---|---|---|---|---|---|---|---|
| **Selection** | | | | | | | | | | | | | |
| Appropriate case definition | $a^1$ 1 star | $a^1$ 1 star | $a^1$ 1 star | $a^1$ 1 star | $a^1$ 1 star | $a^1$ 1 star | $a^1$ 1 star | $a^1$ 1 star | $a^1$ 1 star | $a^1$ 1 star | $a^1$ 1 star | $a^1$ 1 star | $a^1$ 1 star |
| Representativeness of the cases | $a^2$ 1 star | $a^2$ 1 star | $a^2$ 1 star | $a^2$ 1 star | $a^2$ 1 star | $a^2$ 1 star | $a^2$ 1 star | $a^2$ 1 star | $a^2$ 1 star | $a^2$ 1 star | $a^2$ 1 star | $a^2$ 1 star | $a^2$ 1 star |
| Selection of controls | $a^3$ 1 star | $a^3$ 1 star | $a^3$ 1 star | $a^3$ 1 star | $a^3$ 1 star | $a^3$ 1 star | $a^3$ 1 star | $a^3$ 1 star | $b^1$ 0 stars | $b^1$ 0 stars | $b^1$ 0 stars | $b^1$ 0 stars | $b^1$ 0 stars |
| Definition of controls | $a^4$ 1 star | $a^4$ 1 star | $a^4$ 1 star | $a^4$ 1 star | $a^4$ 1 star | $a^4$ 1 star | $a^4$ 1 star | $a^4$ 1 star | $a^4$ 1 star | $a^4$ 1 star | $a^4$ 1 star | $a^4$ 1 star | $a^4$ 1 star |
| **Comparability** | | | | | | | | | | | | | |
| Comparability of cases and controls based on design or analysis | $a^{5,6}$ 2 stars | $a^5$ 1 star | $a^5$ 1 star | $a^5$ 1 star | $a^5$ 1 star | $a^5$ 1 star | $a^5$ 1 star | $a^5$ 1 star | $a^5$ 1 star | $a^5$ 1 star | $a^5$ 1 star | $a^5$ 1 star | $a^5$ 1 star |
| **Exposure** | | | | | | | | | | | | | |
| Exposure assessment | $b^2$ 0 stars | $b^2$ 0 stars | $b^2$ 0 stars | $b^2$ 0 stars | $b^2$ 0 stars | $b^2$ 0 stars | $b^2$ 0 stars | $b^2$ 0 stars | $b^2$ 0 stars | $b^2$ 0 stars | $b^2$ 0 stars | $b^2$ 0 stars | $b^2$ 0 stars |
| Same method of evaluation of cases and controls | $a^7$ 1 star | $a^7$ 1 star | $a^7$ 1 star | $a^7$ 1 star | $a^7$ 1 star | $a^7$ 1 star | $a^7$ 1 star | $a^7$ 1 star | $a^7$ 1 star | $a^7$ 1 star | $a^7$ 1 star | $a^7$ 1 star | $a^7$ 1 star |
| Non-response percentage | $a^8$ 1 star | $a^8$ 1 star | $a^8$ 1 star | $a^8$ 1 star | $a^8$ 1 star | $a^8$ 1 star | $a^8$ 1 star | $a^8$ 1 star | $a^8$ 1 star | $a^8$ 1 star | $a^8$ 1 star | $a^8$ 1 star | $a^8$ 1 star |
| Overall study quality | Good methodological quality | Good methodological quality | Good methodological quality | Good methodological quality | Good methodological quality | Good methodological quality | Good methodological quality | Good methodological quality | Good methodological quality | Good methodological quality | Good methodological quality | Good methodological quality | Good methodological quality |

Note: Assessment using the Newcastle-Ottawa Scale for case-control studies. A study can receive up to one star for each numbered item in the Selection and Exposure categories, and up to two stars for Comparability. $a^1$ The case definition is adequate, with independent validation (1 star); $a^2$ Consecutive or representative cases (1 star); $a^3$ Community controls (1 star); $a^4$ Definition of controls: no history of disease (1 star); $a^5$ The study controls for age (1 star); $a^6$ The study also controls for race (1 star); $a^7$ Same evaluation method as cases and controls: yes (1 star); $a^8$ Same response percentage in both groups (1 star); $b^1$ Hospital controls (0 stars); $b^2$ Exposure assessment: unblinded interviews for cases and controls (0 stars).

**Table 3. Evaluation of risk of bias of cohort studies.**

| Domain | de Morais 2014 [90] | de Rossi 2014 [91] |
|---|---|---|
| Confounding bias | Critical risk of bias | Critical risk of bias |
| Selection bias | Critical risk of bias | Critical risk of bias |
| Bias in the classification of interventions | Moderate risk of bias | Moderate risk of bias |
| Bias due to deviations from planned interventions | Serious risk of bias | Serious risk of bias |
| Bias due to missing data | Low risk of bias | Low risk of bias |
| Bias in measuring results | Moderate risk of bias | Moderate risk of bias |
| Bias in the selection of results reporting | Low risk of bias | Low risk of bias |
| **General bias** | Critical risk of bias | Critical risk of bias |

Note: Assessment using the ROBINS-I tool for cohort studies.

**Table 4. Evaluation of the critical approach case series studies.**

| Question | Elkik 1986 [93] | Bounhoure 2008 [92] |
|---|---|---|
| 1. Were there clear criteria for inclusion in the case series? | No | No |
| 2. Was the disease measured in a standard and reliable way in all participants included in the case series? | Yes | No |
| 3. Were valid methods used to identify the disease in all participants included in the case series? | It is not clear | No |
| 4. Was there a consecutive inclusion of participants in the case series? | No | No |
| 5. Did the case series include all participants? | No | It is not clear |
| 6. Were the demographics of study participants clearly reported? | Yes | Yes |
| 7. Was the clinical information of the participants clearly reported? | Yes | Yes |
| 8. Were the results or results of case follow-up clearly communicated? | Yes | Yes |
| 10. Was the demographic information of the centers or clinics that presented the cases clearly communicated? | No | No |
| 11. Was the statistical analysis appropriate? | Yes | Yes |
| **Critical approach to the study** | Low critical approach | Low critical approach |

Note: Assessment using the Joanna Briggs Institute checklist for case series.

## d. Summary of the results

Outcomes by contraceptive method and study type are presented below. The certainty of evidence assessment is in S13–S14 Appendices and the 2×2 tables for OR calculations are in S15 Appendix. The very low certainty of the evidence in the outcomes was mainly due to the high risk of bias and issues of imprecision.

In case–control studies assessing safety outcomes, exposure was defined as "current use of a specific hormonal contraceptive", and the comparator as "no current use of that specific hormonal contraceptive". This comparator group included both previous and never users. The studies did not report whether participants were simultaneously using non-hormonal contraceptive methods; thus, the comparator group may have included women using non-hormonal methods or no contraception.

### 1. Combined hormonal contraceptives.

#### a. Combined oral contraceptives

The evidence is very uncertain about the effect of COCs on avoiding unwanted pregnancies, ischemic CVD, AMI, VTE, worsening hypertension, weight gain, kidney function, and lipid alterations in hypertensive women; certainty of the evidence: very low. However, COCs may increase hemorrhagic CVD; certainty of evidence: low. Details of COCs outcomes identified through vote counting and narrative synthesis are in S16–S17 Appendices.

#### 1. Unwanted pregnancies

One cohort study [90] assessed 56 hypertensive women (30 exposed to COCs, 26 unexposed), and found no pregnancies in either group.

#### 2. Ischemic CVD

Three case-control studies [76,84,88] analyzed 613 hypertensive women: 299 cases (ischemic CVD; 73 exposed to COCs, 226 unexposed) and 314 controls (no ischemic CVD; 50 exposed, 264 unexposed). One study [84] suggests a possible positive association between COCs and ischemic CVD (crude OR= 3.95 (95% CI = 2.15–7.25), and the others two [76,88] suggest that there may be no difference (crude OR= 1.53 (95% CI = 0.75–3.11) and OR= 0.58 (95% CI = 0.21–1.64)). The certainty of the evidence is very low about the effect of COCs on ischemic CVD.

No meta-analysis was performed given the presence of considerable statistical heterogeneity ($I^2$ = 81%, p value of chi$^2$ = 0.005). Possible sources of heterogeneity in the population were sought: age, degree of arterial hypertension, presence of comorbidities, history of smoking, and management of the underlying pathology; in exposures: definition of exposure (current use), components and concentration of hormones in contraceptives, duration of use of hormonal contraceptives, use of other medications, perfect use compared to typical use of the hormonal contraceptive method and in the outcomes: the way the outcome was determined. It was only possible to perform a subgroup analysis based on the definition of exposure (current use) of contraceptives, since the studies did not describe the aforementioned characteristics in hypertensive women (Fig 2).

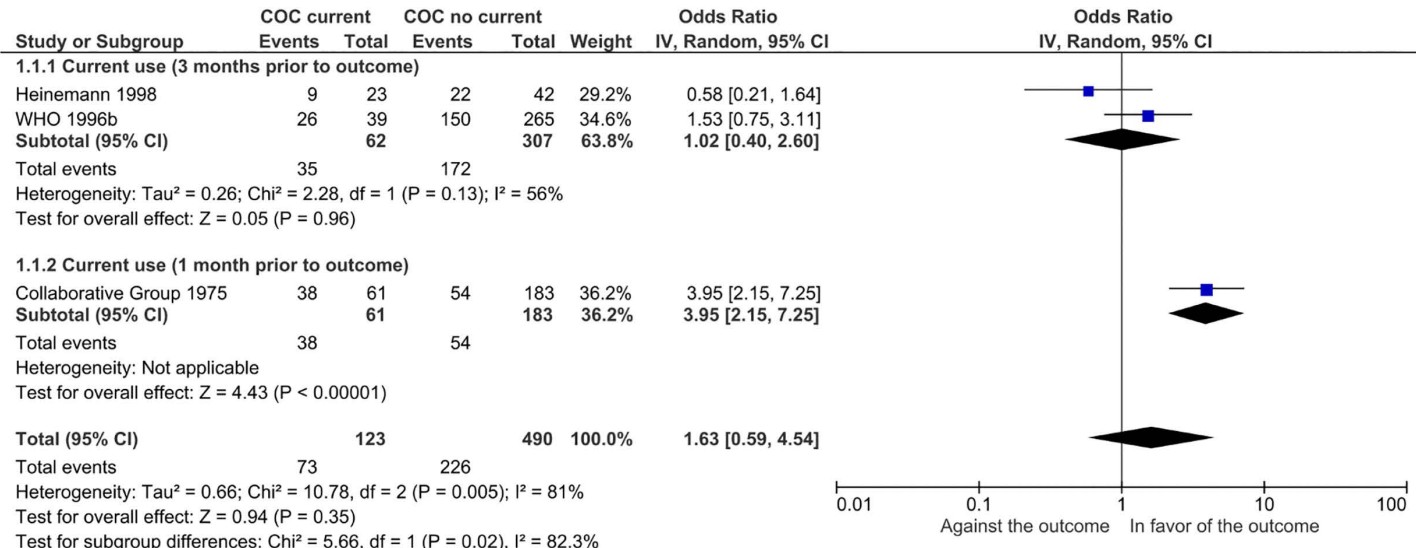

**Fig 2. Subgroup analysis in combined oral contraceptives by exposure definition (outcome: ischemic cerebrovascular event).** Test for subgroup differences: Chi$^2$ = 5.66, df = 1 (P = 0.02), $I^2$ = 82.3%.

The test for subgroup differences indicates a statistically significant effect (p = 0.02), suggesting that COC exposure time may modify its impact on ischemic CVD. However, substantial unexplained heterogeneity exists between the two case-control studies with the same COC exposure time (I² = 56%), making the validity of the effect estimation uncertain due to inconsistency in the studies evaluating this outcome.

### 3. Hemorrhagic CVD

Two case-control studies [84,85] analyzed 839 hypertensive women: 494 cases (hemorrhagic CVD; 79 exposed to COCs, 415 unexposed) and 345 controls (no hemorrhagic CVD; 39 exposed, 306 unexposed). Studies suggest COCs in hypertensive women may increase hemorrhagic CVD (pooled crude OR: 1.64 95% CI 1.08–2.50) (Fig 3). Certainty of evidence: low. The evaluation of publication bias is presented in S18 Appendix.

### 4. AMI

A case-control study [82] and a case-series [92] assessed the presence of AMI and COCs use. The case-control study analyzed 175 hypertensive women: 114 cases (AMI; 27 exposed, 87 unexposed) and 61 controls (no AMI; 6 exposed, 55 unexposed). This study suggests a possible positive association between IAM and COCs (crude OR= 2.85 (95% CI: 1.10–8.93). The case series included 12 users of COCs with AMI, 3 of them were hypertensive. Certainty of the evidence: very low.

### 5. VTE

One case-control study [79] identified 69 cases with a first VTE event and 133 controls. Due to insufficient exposure data, the OR could not be calculated. Certainty of evidence: very low.

### 6. Worsening of baseline condition

A cohort study [90] evaluated systolic (SBP) and diastolic blood pressure (DBP) changes in 56 hypertensive women (30 exposed, 26 unexposed) and found no significant differences at six months between COC users and non-users. Users (n = 30)= mean SBP±Standard error(SE)= 127.8 ± 2.1 mmHg to 126.6 ± 2.5 mmHg, p = 0.57; mean DBP ± SE = 83.9 ± 1.3 mmHg to 83.7 ± 1.8 mmHg, p = 0.93. Non-users (n = 26)= SBP ± SE = 129.0 ± 2.5 mmHg to 130.3 ± 2.4 mmHg, p = 0.70; DBP ± SE = 87.6 ± 1.9 mmHg to 87.0 ± 1.4 mmHg, p = 0.57. Another cohort study [105] analyzed 65 hypertensive women (40 exposed, 25 unexposed) and found no changes in daytime SBP or nocturnal-daytime DBP at six months. However, nocturnal SBP decreased in COC users (114.5 ± 1.79 mmHg to 110.2 ± 1.77 mmHg, p = 0.032) but remained unchanged in non-users. Certainty of evidence: very low.

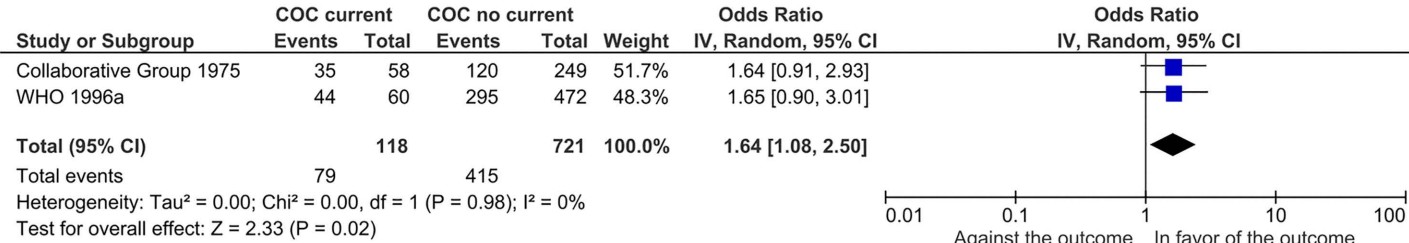

**Fig 3. Current use of combined oral contraceptives compared with no current use (past use or never used) in hypertensive women. Outcome: hemorrhagic cerebrovascular event.** Heterogeneity: Tau²=0.00; Chi²=0.00, df=1 (P=0.98); I²=0%.

### 7. Increase of weight

**Body Mass Index (BMI):** Two cohort studies [90,91] assessed BMI in 121 hypertensive women (70 exposed, 51 unexposed). A study [90] reported a BMI decrease in COCs users after six months (mean BMI±SE: initial=30.3±0.9 Kg/m² to 29.8±0.9 Kg/m², p=0.04), with no changes in non-users. The other study [91] found no BMI differences in either group after six months (users: mean BMI+-SE: initial=28.9+−0.78 Kg/m² to 28.45+−0.79 Kg/m² (p=0.620); non-users: initial=30.79+−1.20 Kg/m² to 29.01+−1.11 Kg/m2 (p=0.27)). Certainty of evidence: very low.

**Abdominal circumference:** One cohort study [90] evaluated abdominal circumference in 56 hypertensive women and found no significant changes after six months (users: mean abdominal perimeter±SE=97.9±2.0 cm to 97.2±2.0 cm, p=0.66; non-users: 95.9±2.6 cm to 94.8±2.6 cm, p=0.76). Certainty of evidence: very low.

### 8. Kidney function

One cohort study [90] found no significant differences in serum creatinine levels between COCs users and non-users after six months (users: mean creatinine±SE=0.8±0.0 mg/dL to 0.7±0.0 mg/dL, p=0.33; non-users=0.8±0.0 mg/dL to 0.7±0.0 mg/dL, p=0.41). Certainty of evidence: very low.

### 9. Lipid profile

Two cohort studies [90,91] evaluated lipid profile changes in a total of 121 hypertensive women (70 exposed, 51 unexposed) over six months. No significant differences were observed between COC users and non-users in total cholesterol, LDL cholesterol, HDL cholesterol, or triglyceride levels. Certainty of evidence: very low.

### b. Combined vaginal ring

The certainty of the evidence is very low about the effect of the vaginal ring on worsening hypertension, and lipid level alterations in hypertensive women; certainty of evidence: very low. A single case series [93] assessed these outcomes in 12 hypertensive women using the combined vaginal ring. S19–S20 Appendices present the narrative synthesis of quantitative outcomes for the vaginal ring.

### 1. Worsening of hypertension

No significant changes in SBP or DBP were observed over 12 months in women using the vaginal ring. Values remained stable across control, treatment, and recovery cycles (p value for DBP and SBP at each measurement time compared to cycle 0: not significant (the significance value is not reported)).

### 2. Lipids

For 12 months of observation of the women with the vaginal ring, a decrease in total cholesterol was observed (mean±Standard deviation(SD): 182±30 mg/100mL to 157±26 mg/100mL, p<0.01), driven by a reduction in HDL cholesterol (59±10 mg/100mL to 38±3 mg/100mL, p<0.001), with no significant changes in LDL cholesterol. Triglycerides also decreased (52±17 mg/100mL to 38±11 mg/100mL, p<0.05).

### c. Combined injectable contraceptives

The evidence is very uncertain about the effect of combined injectable contraceptives on CVD, AMI, and VTE in hypertensive women; certainty of evidence: very low. A case-control study [80] assessed these outcomes. S21 Appendix presents the synthesis of results for combined injectable contraceptives using the vote-counting method.

### 1. Hemorrhagic and ischemic CVD

The study analyzed 942 hypertensive women: 573 cases (CVD; 2 exposed, 571 unexposed) and 369 controls (no CVD; 1 exposed, 368 unexposed) (crude OR= 1.29 (IC 95% 0.07–76.24)).

2. **AMI**

The study analyzed 138 hypertensive women: 85 cases (AMI; 1 exposed, 84 unexposed) and 53 controls (no AMI; 0 exposed, 53 unexposed) (crude OR = 1.26; 95% CI: 0.04–38.27).

3. **VTE**

The analyzed 132 hypertensive women: 41 cases (VTE; 0 exposed, 41 unexposed) and 91 controls (no VTE; 0 exposed, 91 unexposed).

 **2. Progestin-only contraceptives.**

a. **Progestin-only pill**

There is very low certainty in the evidence about the effect of progestin-only pill (POPs) on CVD, AMI, and VTE in hypertensive women; certainty of evidence: very low. A case-control study [80], assessed these outcomes for this contraceptive. S22 Appendix presents the synthesis of results for POPs using the vote-counting method.

1. **Hemorrhagic and ischemic CVD**

This study included 960 hypertensive women (585 cases (14 exposed to POPs, 571 non-exposed), 375 controls (7 exposed to POPs, 368 non-exposed)). The study suggested that POPs may reduce, increase or have little to no effect on CVD in hypertensive women (crude OR= 1.29 (IC 95% 0.48–3.81)), but the evidence is very uncertain.

2. **AMI**

For AMI, they included 139 hypertensive women (85 cases (1 exposed to POPs, 84 non-exposed), 54 controls (1 exposed to POPs, 53 non-exposed)). The study suggested that POPs may reduce, increase or have little to no effect on AMI in hypertensive women (crude OR= 0.63 (IC 95% 0.01–50)), but the certainty of the evidence is very low.

3. **VTE**

The study included 135 hypertensive women: 42 cases (VTE; 1 exposed, 41 unexposed) and 93 controls (no VTE; 2 exposed, 91 unexposed). The study suggested that POPs may reduce, increase or have little to no effect on VTE in hypertensive women (crude OR=1.11 (IC 95% 0.02–21.86)), but the evidence is very uncertain.

b. **Progestin-only injectable**

The certainty of the evidence is very low about the effect of progestin-only injectable on CVD, AMI, and VTE in hypertensive women; certainty of evidence: very low. A case-control study [80], assessed these outcomes for this contraceptive. S23 Appendix presents the details of the outcomes identified using the vote-counting method for progestin-only injectables.

1. **Hemorrhagic and ischemic CVD**

The study included 944 hypertensive women: 576 cases (CVD; 5 exposed, 571 unexposed) and 368 controls (no CVD; 0 exposed, 368 unexposed). The study suggested that progestin-only injectables may reduce, increase or have little to no effect on CVD in hypertensive women (crude OR= 6.45 (IC 95% 0.35–118.3)), but the evidence is very uncertain.

2. **AMI**

The study included 137 hypertensive women: 84 cases (AMI; 0 exposed, 84 unexposed) and 53 controls (no AMI; 0 exposed, 53 unexposed). Progestin-only injectables may reduce, increase or have little to no effect on AMI, but there is very low certainty in the evidence.

3. **VTE**

The study included 133 hypertensive women: 41 cases (VTE; 0 exposed, 41 unexposed) and 92 controls (no VTE; 1 exposed, 91 unexposed). The study suggested that progestin-only injectables may reduce, increase or have little to no effect on hemorrhagic or ischemic CVD in hypertensive women (crude OR= 1.11 (IC95% 0.04–33.74)), but the evidence is very uncertain.

 **3. Combined hormonal and progestin-only contraceptives.** The certainty of the evidence is very low about the effect of COCs and POPs on ischemic or hemorrhagic CVD, AMI and peripheral arterial disease in hypertensive women; certainty of evidence: very low. The synthesis of the results using the voting count method is presented in S24 Appendix.

a. **Ischemic CVD**

Two case-control studies [86,87,89], evaluated the presence of ischemic CVD among hypertensive women users of COC or POP. No meta-analysis was performed given the presence of considerable statistical heterogeneity ($I^2 = 80\%$, p value of the $chi^2 = 0.03$). Only a subgroup analysis could be performed based on the definition of exposure (current use) of contraceptives used by women in the studies (Fig 4).

 The test of subgroup differences suggests there is a statistically significant subgroup effect (p = 0.03). This suggests the exposure time of COCs or POPs could modify their effect in the presence of ischemic CVD. The evidence is very uncertain on the effect of COCs or POPs on ischemic CVD in hypertensive women.

b. **Hemorrhagic and ischemic CVD**

A case-control study [78] included 117 hypertensive women: 51 cases (CVD; 21 exposed, 30 unexposed) and 66 controls (no CVD; 23 exposed, 43 unexposed). This study suggests that the use of COCs or POPs may reduce, increase or have little to no effect on CVD in hypertensive women (crude OR= 1.31 (95% CI 0.62–2.78)), but the certainty of the evidence is very low.

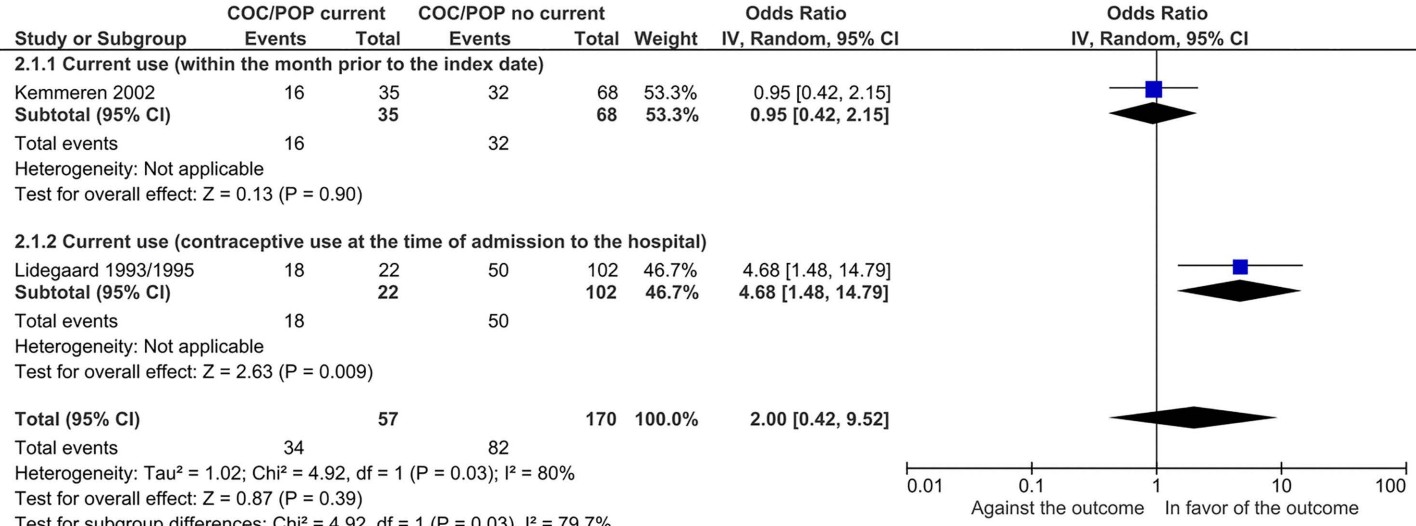

**Fig 4. Subgroup analysis in combined or progestin-only oral contraceptives by exposure definition (outcome: ischemic cerebrovascular event).** Test for subgroup differences: $Chi^2 = 4.92$, df = 1 (P = 0.03), $I^2 = 79.7\%$.

c. **AMI**

Two case-control studies [77,83] assessed AMI in hypertensive women, comparing cases (AMI) and controls (no AMI) based on COC or POP use. The evidence is very uncertain about the effect of COC or POP on AMI in hypertensive women (crude OR= 1.15 (95% CI 0.60–2.19)). Certainty of the evidence: very low (Fig 5). The evaluation of publication bias is presented in S25 Appendix.

d. **Peripheral arterial disease**

A case-control study [81] analyzed 98 hypertensive women: 43 cases (peripheral arterial disease; 16 exposed, 27 unexposed) and 55 controls (no peripheral arterial disease; 19 exposed, 36 unexposed). The study suggested that COCs or POPs may reduce, increase or have little to no effect on peripheral arterial disease in hypertensive women (crude OR= 1.12 (95% CI 0.45–2.79)), but the evidence is very uncertain.

The presentation summary of this article is in S26 Appendix. Articles screened by title and abstract from the databases can be found in S27 Appendix.

**4. Emergency contraceptives and levonorgestrel-releasing intrauterine devices.** No studies were identified that evaluated the safety or effectiveness of emergency contraceptives or levonorgestrel-releasing intrauterine devices in women with hypertension.

## 4. Discussion

### a. Findings and interpretation

Evidence suggests that COCs may increase hemorrhagic CVD in hypertensive women, although the certainty of the evidence is low. The evidence is very uncertain regarding the effect of COCs on pregnancy prevention, ischemic CVD, AMI, venous thromboembolism, worsening hypertension, weight gain, kidney function, and lipid profile alterations in hypertensive women. The effect of COCs or POPs on CVD, AMI, and peripheral arterial disease is also very uncertain, as is the effect of POPs, progestin-only injectables, or combined injectable contraceptives on CVD, AMI, and venous thromboembolism, and the combined vaginal ring on worsening hypertension, and lipid profile alteration. The very low certainty of the evidence was mainly due to a high risk of bias and issues of imprecision. For outcomes with considerable heterogeneity, it was not possible to identify its sources.

We found no studies assessing the safety or effectiveness of the patch, implants, emergency contraceptives, or levonorgestrel-releasing intrauterine devices (LNG-IUDs) in hypertensive women. PID, vaginal infections, fertility loss, liver function alterations, and the Pearl Index were not found.

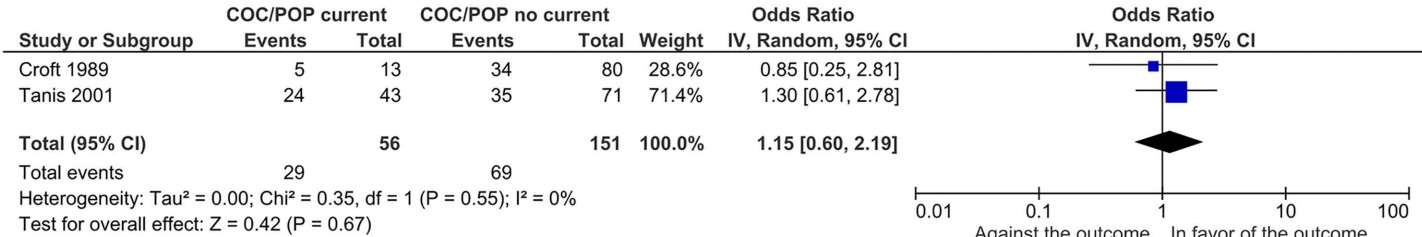

**Fig 5. Current use of combined oral contraceptives or progestin-only pills compared with no current use (past use or never used) of combined oral contraceptives or progestin-only pills in hypertensive women.** Outcome: acute myocardial infarction. Heterogeneity: Tau² = 0.00; Chi² = 0.35, df = 1 (P = 0.55); I² = 0%.

 

### b. Results in the context of what is known

WHO recommendations on the use of COCs in hypertensive women [8], are based on primary studies, systematic reviews not specifically focused on hypertensive women [94,95] and expert consensus. The individual studies are found in two reviews (critically low quality (AMSTAR-2)) from 2002 and 2006 assessing COC safety in hypertensive women [9,10], concluding that COC users have a higher risk of CVD and AMI compared to non-users, with no increase in venous thromboembolism. Differences with our results may relate to the certainty assessments, precision evaluation based on optimal data sizes (unclear in those reviews), and exclusion of ineligible studies [29,62–69,96].

WHO lacks supporting evidence for the combined patch, vaginal ring, and combined injectables, but classifies all combined hormonal contraceptives as category 3–4 in the Medical Eligibility Criteria for Contraceptive Use (MEC), indicating risks generally outweigh benefits [8]. We identified one case series on the vaginal ring [93] and one case-control study on combined injectables [80], both providing very low certainty of the evidence for the outcomes evaluated.

WHO recommendations [8] for progestogen-only contraceptives rely on a single study [80], the only one found for POPs and injectables. Similarly, recommendations for implants, emergency contraceptives, and LNG-IUDs lack supporting studies, consistent with our findings. WHO classifies progestogen-only contraceptives as category 1–3 in the MEC, indicating no restriction or that risks outweigh benefits [8]. In our review, the certainty of evidence for the outcomes was very low. The characteristics and mechanisms of action of hormonal contraceptives are provided in S1 Appendix [97–102].

### c. Clinical implications

Evidence suggests that COCs in hypertensive women may increase hemorrhagic CVD, but the certainty is low. For other outcomes, the certainty is very low; therefore, the safety and effectiveness of COCs, the vaginal ring, combined injectables, and progestin-only methods in hypertensive women cannot be conclusively established.

### d. Research implications

The ideal study to assess the safety and effectiveness of hormonal contraception in hypertensive women is a pragmatic clinical trial. Cohort studies using administrative data and causal inference analyses also represent viable alternatives. Upcoming research should be methodologically rigorous, with adequate statistical power and strategies to reduce bias.

### e. Strengths and Limitations

Our review comprehensively searched for studies on hormonal contraceptives in hypertensive women, including various study designs. Unlike previous reviews focused only on COCs, we evaluated all hormonal contraceptive methods. Risk of bias and study quality were assessed using tools specific to each design [19–21], and we applied GRADE to evaluate certainty of evidence. We synthesized data qualitatively and quantitatively. All processes were documented in a protocol registered in PROSPERO [22].

Limitations include the inability to evaluate hypertension severity and contraceptive safety due to insufficient classification in the studies. All outcomes had very low certainty of evidence, except for the possible association between COCs and hemorrhagic CVD (low evidence). There are no adequate tools to assess bias in case-control studies or case series. Information on comorbidities, cardiovascular risk factors, duration of contraceptive exposure, and ages of hypertensive women using hormonal contraceptives was unavailable in the studies.

No studies evaluated outcomes for the patch, levonorgestrel and etonogestrel implants, emergency contraceptives, or LNG-IUDs.

## 5. Conclusions

This systematic review suggests that COCs may increase hemorrhagic CVD in hypertensive women, although the certainty of the evidence is low. For all other outcomes evaluated, including those related to COCs, vaginal ring, combined

injectables, and progestin-only methods, the certainty of the evidence was very low. The safety and effectiveness of hormonal contraceptive methods in hypertensive women of reproductive age remain uncertain.

There is a need for high-quality primary studies in this population. Future research should aim to reduce bias, include adequately powered sample sizes, and clearly define study populations. Given the substantial uncertainty, existing clinical recommendations should be maintained, with particular caution regarding the use of COCs due to their possible association with hemorrhagic CVD.

## Supporting information

**S1 Appendix. Classification of hormonal contraceptive methods, percentages of unwanted pregnancies with perfect and typical use during the first year and side effects and side effects.**
(PDF)

**S2 Appendix. Classification of non-hormonal contraceptive methods, percentage of unwanted pregnancies in the first year according to the hormonal contraceptive method with perfect and typical use.**
(PDF)

**S3 Appendix. PRISMA 2020 checklist.**
(PDF)

**S4 Appendix. PRISMA 2020 for abstracts checklist.**
(PDF)

**S5 Appendix. MOOSE checklist for meta-analyses of observational studies.**
(PDF)

**S6 Appendix. Sources of information.**
(PDF)

**S7 Appendix. Search logs.**
(PDF)

**S8 Appendix. Characteristics of studies not included.**
(PDF)

**S9 Appendix. Data extraction format for studies on hormonal contraceptives and hypertensive women.**
(PDF)

**S10 Appendix. Diagram for data synthesis.**
(PDF)

**S11 Appendix. Detailed characteristics of the included studies and assessment of risk of bias, methodological quality and critical approach.**
(PDF)

**S12 Appendix. General characteristics of the exposures/interventions and study comparators.**
(PDF)

**S13 Appendix. Summary table of results of included GRADE studies.**
(PDF)

**S14 Appendix. Evidence quality assessment table for individual studies.**
(PDF)

**S15 Appendix. 2x2 tables of the included case-control studies.**
(PDF)

**S16 Appendix. Synthesis of outcomes related to the use of combined oral contraceptives using the vote counting method.**
(PDF)

**S17 Appendix. Narrative synthesis of continuous outcomes for combined oral contraceptives.**
(PDF)

**S18 Appendix. ROB-ME Current use of combined oral contraceptives compared with no current use (past use or never used) of combined oral contraceptives in hypertensive women.** Outcome: hemorrhagic cerebrovascular event.
(PDF)

**S19 Appendix. Synthesis of results related to the use of the combined contraceptive vaginal ring using the vote counting method.**
(PDF)

**S20 Appendix. Narrative synthesis of continuous outcomes for vaginal ring.**
(PDF)

**S21 Appendix. Synthesis of the results related to the use of the combined injectable contraceptive using vote-counting method.**
(PDF)

**S22 Appendix. Synthesis of results related to the use of progestin-only pills using the vote counting method.**
(PDF)

**S23 Appendix. Synthesis of results related to the use of progestin-only injectables using the vote counting method.**
(PDF)

**S24 Appendix. Synthesis of results related to the use of combined oral contraceptives or progestin-only pills using the vote counting method.**
(PDF)

**S25 Appendix. ROB-ME Current use of combined oral contraceptives or progestin-only pills compared with no current use (past use or never used) of combined oral contraceptives or progestin-only pills in hypertensive women.** Outcome: acute myocardial infarction.
(PDF)

**S26 Appendix. Presentation summary.**
(PDF)

**S27 Appendix. Articles screened by title and abstract from the databases.**
(PDF)

## Acknowledgments

Thanks to the Universidad Nacional de Colombia for allowing the academic growth of its students, as well as the development of this study.

## Author contributions

**Conceptualization:** Natalia Losada-Trujillo, Kelly Estrada-Orozco, Álvaro Javier Burgos-Cárdenas, Hernando Gaitán-Duarte.

**Data curation:** Natalia Losada-Trujillo, Oscar Julián Velasco-Lancheros, Brahiam Alejandro Ramírez-Vargas, Álvaro Javier Burgos-Cárdenas, Paula Andrea González-Caicedo, María José Hoyos Bedoya.

**Formal analysis:** Natalia Losada-Trujillo, Kelly Estrada-Orozco, Álvaro Javier Burgos-Cárdenas, Hernando Gaitán-Duarte.

**Investigation:** Natalia Losada-Trujillo, Kelly Estrada-Orozco, Oscar Julián Velasco-Lancheros, Brahiam Alejandro Ramírez-Vargas, Álvaro Javier Burgos-Cárdenas, Paula Andrea González-Caicedo, María José Hoyos Bedoya, Hernando Gaitán-Duarte.

**Methodology:** Natalia Losada-Trujillo, Kelly Estrada-Orozco, Oscar Julián Velasco-Lancheros, Brahiam Alejandro Ramírez-Vargas, Álvaro Javier Burgos-Cárdenas, Hernando Gaitán-Duarte.

**Resources:** Natalia Losada-Trujillo.

**Software:** Natalia Losada-Trujillo.

**Supervision:** Kelly Estrada-Orozco, Álvaro Javier Burgos-Cárdenas, Hernando Gaitán-Duarte.

**Validation:** Kelly Estrada-Orozco, Álvaro Javier Burgos-Cárdenas, Hernando Gaitán-Duarte.

**Visualization:** Natalia Losada-Trujillo, Oscar Julián Velasco-Lancheros, Brahiam Alejandro Ramírez-Vargas.

**Writing – original draft:** Natalia Losada-Trujillo, Oscar Julián Velasco-Lancheros, Brahiam Alejandro Ramírez-Vargas.

**Writing – review & editing:** Natalia Losada-Trujillo, Kelly Estrada-Orozco, Oscar Julián Velasco-Lancheros, Brahiam Alejandro Ramírez-Vargas, Hernando Gaitán-Duarte.

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
