## [Editor Report · Decision Letter 0]

27 Aug 2025

Dear Dr. LOSADA TRUJILLO,

Thank you for submitting your manuscript to PLOS ONE. After careful consideration, we feel that it has merit but does not fully meet PLOS ONE’s publication criteria as it currently stands. Therefore, we invite you to submit a revised version of the manuscript that addresses the points raised during the review process.

We look forward to receiving your revised manuscript.

Kind regards,

Kwang-Sig Lee

Academic Editor

PLOS ONE

Journal Requirements:

https://journals.plos.org/plosone/s/file?id=ba62/PLOSOne_formatting_sample_title_authors_affiliations.pdf ..

2. Please amend your manuscript to include your abstract after the title page.

Additional Editor Comments:

I would like to point out that all three invited reviewers declined to review this manuscript in large part because Table 1 and Table 2 look crummy (at least in the final pdf version). I would like to ask the authors to revise Table 1 and Table 2 in a clean format.

While revising your submission, please upload your figure files to the Preflight Analysis and Conversion Engine (PACE) digital diagnostic tool, https://pacev2.apexcovantage.com/ . PACE helps ensure that figures meet PLOS requirements. To use PACE, you must first register as a user. Registration is free. Then, login and navigate to the UPLOAD tab, where you will find detailed instructions on how to use the tool. If you encounter any issues or have any questions when using PACE, please email PLOS at . PACE helps ensure that figures meet PLOS requirements. To use PACE, you must first register as a user. Registration is free. Then, login and navigate to the UPLOAD tab, where you will find detailed instructions on how to use the tool. If you encounter any issues or have any questions when using PACE, please email PLOS at figures@plos.org . Please note that Supporting Information files do not need this step.. Please note that Supporting Information files do not need this step.

---

## [Author Response · Author response to Decision Letter 1]

13 Sep 2025

Dear Editor and Reviewers,

We sincerely thank you for the careful review of our manuscript, entitled: "Safety and effectiveness of hormonal vs non-hormonal or no contraception in women with hypertension and future fertility desire: a broad-scope systematic review". We appreciate the insightful comments and constructive feedback.

Below, we provide a detailed, point by point response to the comments. All changes have been incorporated into the manuscript, and we have carefully reformatted the tables to enhance readability and consistency.

• Tables: All tables, both within the main manuscript and the supplementary material, have been carefully reformatted to enhance their visual presentation, alignment, and clarity. We standardized font size, optimized spacing, and table legends to comply with journal policies. Additionally, we erased the colour used in subheadings to ensure a more fluent and consistent reading.

We sincerely thank you once again for your valuable time, thoughtful comments, and constructive suggestions, which have significantly improved our manuscript. We greatly appreciate the opportunity to review our work and hope the changes made fully address the concerns raised.

Sincerely,

Natalia Losada-Trujillo on behalf of all co-authors

---

## [Decision Letter · Decision Letter 1]

2 Feb 2026

Dear Dr. LOSADA TRUJILLO,

Thank you for submitting your manuscript to PLOS ONE. After careful consideration, we feel that it has merit but does not fully meet PLOS ONE’s publication criteria as it currently stands. Therefore, we invite you to submit a revised version of the manuscript that addresses the points raised during the review process.

As the major revision is marked by one of the reviewer on the point that either the control is women using non hormonal contraceptive or women using no contraceptive. It should be clear and justified.

The second review comments that there need to specify the specific rule for the vote count (e.g., direction of effect based on point estimate, statistical significance).

The statement by the authors that the evidence is uncertain should also replace with suitable synonyms.

Table 1 formatting needs correction so that it could be easily scanned for other researcher as reference.

Key columns should include: Study, Design, Population (n, age, % hypertensive), Intervention/Exposure, Comparator, and Outcomes Assessed.

Figures 2-5: I² statistic for heterogeneity should included in the figure caption or within the plot itself for immediate reference.

We look forward to receiving your revised manuscript.

Kind regards,

Noor Jahan

Academic Editor

PLOS One

Journal Requirements:

Additional Editor Comments :

As the major revision is marked by one of the reviewer on the point that either the control is women using non hormonal contraceptive or women using no contraceptive. It should be clear and justified.

The second review comments that there need to specify the specific rule for the vote count (e.g., direction of effect based on point estimate, statistical significance).

The statement by the authors that the evidence is uncertain should also replace with suitable synonyms.

Table 1 formatting needs correction so that it could be easily scanned for other researcher as reference.

Key columns should include: Study, Design, Population (n, age, % hypertensive), Intervention/Exposure, Comparator, and Outcomes Assessed.

Figures 2-5: I² statistic for heterogeneity should included in the figure caption or within the plot itself for immediate reference.

Reviewers' comments:

Reviewer's Responses to Questions

**Comments to the Author**

Reviewer #1: (No Response)

Reviewer #2: All comments have been addressed

2. Is the manuscript technically sound, and do the data support the conclusions?

Reviewer #1: No

Reviewer #2: Yes

3. Has the statistical analysis been performed appropriately and rigorously?

Reviewer #1: Yes

Reviewer #2: Yes

4. Have the authors made all data underlying the findings in their manuscript fully available?

Reviewer #1: Yes

Reviewer #2: Yes

5. Is the manuscript presented in an intelligible fashion and written in standard English?

Reviewer #1: Yes

Reviewer #2: Yes

Reviewer #1: I cannot comment if previous comments addressed as I am reviewing this manuscript for the 1st time. One basic limitation is the studies compare women using hormonal contraceptives and women not using hormonal contraceptives. The do not compare to women not using any contraceptive, so the women maybe using barrier methods or IUCD etc. So it is not possible to comment or deduct that efficacy of hormonal contraceptives in preventing pregnancy is very low. It should be clarified what the control group is

Reviewer #2: For clarity, please specify the specific rule used for the vote count (e.g., direction of effect based on point estimate, statistical significance). A brief sentence in the methods or a reference to the approach would suffice. The repeated phrase "the evidence is very uncertain" is accurate but can become repetitive. Consider occasionally using synonyms like "the evidence is of very low certainty" or "the estimated effect is very imprecise" for stylistic variation, while maintaining the central GRADE message. Table 1 (General Characteristics): In the provided draft, this table's formatting is disrupted. Please ensure the final version presents this information in a clear, columnar format that is easy for readers to scan. Key columns should include: Study, Design, Population (n, age, % hypertensive), Intervention/Exposure, Comparator, and Outcomes Assessed.

Figures 2-5 (Forest Plots): These are essential and well-presented. A minor enhancement would be to ensure the I² statistic for heterogeneity is included in the figure caption or within the plot itself for immediate reference.

**Do you want your identity to be public for this peer review?** For information about this choice, including consent withdrawal, please see our For information about this choice, including consent withdrawal, please see our Privacy Policy .

Reviewer #1: No

Reviewer #2: **Yes:** Noor KamilNoor Kamil

---

## [Author Response · Author response to Decision Letter 2]

7 Mar 2026

Response to Reviewers

Dear Prof. Noor Jahan and Reviewers,

We sincerely appreciate the time and care devoted to evaluating our manuscript, as well as your valuable observations. We have addressed each of the comments received and believe that the revisions and adjustments made have strengthened our manuscript. Below, we present a detailed point-by-point response to each comment, indicating the specific location of the changes in the manuscript.

1. Reviewer #1 comments:

a. Comment 1.1: Clarification of control group composition

Reviewer's comment: “One basic limitation is the studies compare women using hormonal contraceptives and women not using hormonal contraceptives. The do not compare to women not using any contraceptive, so the women maybe using barrier methods or IUCD etc. So it is not possible to comment or deduct that efficacy of hormonal contraceptives in preventing pregnancy is very low. It should be clarified what the control group is”.

Response:

We sincerely appreciate this observation. To provide the greatest possible clarity, we would like to address this comment comprehensively by separating our response into two main aspects:

a. Safety of hormonal contraceptives

For this systematic review, in order to evaluate the safety of contraceptive methods, our eligibility criteria allowed the inclusion of randomized controlled trials (RCTs), cohort studies, case-control studies, case reports, case series, clinical trial registries, and adverse event reports. This decision was driven by the need to capture rare adverse events, which might be underrepresented in clinical trials. In contrast, for the evaluation of effectiveness, we prioritized the inclusion of RCTs and cohort studies (see Methods section, Eligibility criteria). After study selection, we identified case-control studies, cohort studies, and case series for safety outcomes, and one cohort study (de Morais 2014) to evaluate the effectiveness of combined oral contraceptives (Table 1).

Our research question aimed to evaluate the safety and effectiveness of hormonal contraceptive methods (intervention/exposure) versus non-hormonal contraceptive methods or non-use of contraceptive methods (comparator). However, in the case-control studies identified—which were used to evaluate safety—and given their inherent methodological nature, researchers identified women who experienced major adverse cardiovascular events, peripheral arterial disease, or venous thromboembolism (cases) and matched them with women who had not experienced these outcomes (controls). Subsequently, questionnaires were administered to all participants to assess their prior and current exposures to contraceptive methods. In these studies, exposure was reported as “current use of hormonal contraceptives” (according to the type of contraceptive, as detailed in Table 1), and the comparator was defined as “no current use of hormonal contraceptives”, which encompassed both prior use and never use of hormonal contraceptive methods. Unfortunately, these studies did not collect information on whether women in the “no current use of hormonal contraceptives” group were using non-hormonal methods or were not using any type of contraception at the time of exposure measurement. Therefore, throughout the manuscript, we specify that in these case-control studies, the exposure by type of contraceptive, as declared by the studies' authors, is “current use of hormonal contraceptives” compared with “no current use of hormonal contraceptives”, recognizing that this latter group could include women using non-hormonal methods or no contraception. This “no current use” group represents our comparison or reference group—women without increased risk attributable to hormonal contraceptives in relation to the safety outcomes evaluated.

We recognize this as an inherent limitation of the available published literature. Unfortunately, we do not have access to the raw data from these studies to perform additional analyses; we can only work with the data as reported by the investigators in their original studies. Recognizing that this exposure-comparator definition may cause confusion among readers, we have added a clarification in the Results section (Summary of the results) to explicitly address this important consideration in the identified case-control studies, so that the composition of the control group for safety outcomes is clear. We thank the reviewer and the editor again for their appreciation.

b. Effectiveness of hormonal contraceptives

We thank the reviewer for pointing out the limitation related to contraceptive effectiveness when commenting “So it is not possible to comment or deduct that efficacy of hormonal contraceptives in preventing pregnancy is very low”. After carefully reconsidering our data, we have completely removed the effectiveness outcome of the combined vaginal ring from the manuscript, considering that this is an outcome reported in a case series (Elkik 1986), which is a design we had not considered for evaluating contraceptive effectiveness, having prioritized RCTs and cohort studies for these outcomes. The Elkik 1986 study is maintained only for the evaluation of safety outcomes related to the use of combined vaginal ring, where its methodological design is appropriate and presents no changes.

However, for combined oral contraceptives, in relation to effectiveness (prevention of unwanted pregnancies), the identified cohort study (Morais 2014) compared the effectiveness of combined oral contraceptives (exposure) versus use of condoms or copper intrauterine device (comparator) (Table 1). When evaluating the quality of evidence from this study (Appendix S14), we identified that this study presented extremely serious risk of bias. Using the ROBINS-I tool, we found: critical risk of confounding bias (results were not adjusted for possible confounding variables, with risk of residual confounding) and selection bias (assignment to intervention or comparator group may have been related to participant characteristics); serious risk of bias due to deviations from planned interventions (investigators and participants were aware of assignment to intervention/comparator, increasing susceptibility to this bias); moderate risk of bias in the classification of interventions (choice of intervention/comparator group may have been affected by knowledge of the outcomes or risk of the outcomes) and in outcome measurement (outcome measurement may have been influenced by knowledge of the type of contraceptive received); low risk of bias due to missing data and selective reporting of results. The study also presented serious precision problems (the optimal information size could not be calculated with the data provided by the authors). Consequently, the certainty of the evidence was rated as very low (Appendix S14).

In the original manuscript, we stated “the evidence is very uncertain regarding the effect of COCs and combined vaginal ring on pregnancy prevention”. Following the reviewer's valuable observation, we have completely removed any mention of the effectiveness of the combined vaginal ring, as it is not an appropriate study to evaluate this outcome. We maintain only the statement about the uncertainty of the evidence for combined oral contraceptives in relation to the prevention of unwanted pregnancies in hypertensive women, based on the Morais 2014 study. We thank the reviewer again for this observation.

Specific changes:

• In the Results section (Summary of the results, page 24, lines 321-326), a clarification was added regarding the control group of case-control studies evaluating the safety of hormonal contraceptive methods in hypertensive women: “In case–control studies assessing safety outcomes, exposure was defined as “current use of a specific hormonal contraceptive”, and the comparator as “no current use of that specific hormonal contraceptive”. This comparator group included both previous and never users. The studies did not report whether participants were simultaneously using non-hormonal contraceptive methods; thus, the comparator group may have included women using non-hormonal methods or no contraception”.

• Results section, page 29, lines 432-434: We eliminated the effectiveness data for the vaginal ring (Elkik 1986) and any reference to this study in the context of contraceptive effectiveness. This study is maintained only for safety outcomes, where its methodological design is appropriate. Additionally, we removed from Appendix S14, page 174, the quality assessment of the unwanted pregnancies outcome from the Elkik 1986 study.

• Throughout the manuscript: We carefully reviewed all text to eliminate any statements about effectiveness in relation to this case series.

2. Reviewer #2 comments:

a. Comment 2.1: Vote count methodology specification

Reviewer's comment: “For clarity, please specify the specific rule used for the vote count (e.g., direction of effect based on point estimate, statistical significance). A brief sentence in the methods or a reference to the approach would suffice”.

Response:

Thank you for this methodological clarification request. We have now explicitly described our vote counting rules in the Methods section (Data synthesis).

Specific changes:

• Methods section, page 12, lines 239-243: Added new paragraph explaining how we performed vote counting: "For all outcomes in which meta-analysis could not be performed, synthesis was performed by vote counting based on the direction of effect (25). Each effect estimate was classified as favoring the outcome, against the outcome, or showing no difference. Vote counting consisted of tallying the number of studies contributing to each direction of effect for a given outcome and was interpreted in conjunction with the certainty of the evidence".

• The reference presenting the other statistical synthesis methods proposed by Cochrane, which include vote counting, was added.

b. Comment 2.2: Stylistic variation for certainty language

Reviewer's comment: “The repeated phrase "the evidence is very uncertain" is accurate but can become repetitive. Consider occasionally using synonyms like "the evidence is of very low certainty" or "the estimated effect is very imprecise" for stylistic variation, while maintaining the central GRADE message”.

Response:

We agree that stylistic variation improves readability while maintaining the essential GRADE message about evidence certainty. We have replaced repetitive uses of "the evidence is very uncertain" with appropriate synonyms throughout the Results and Discussion sections.

Specific changes:

• We used different synonyms in the Results section for “the evidence is very uncertain about…”, such as “The certainty of the evidence is very low” or “There is very low certainty in the evidence about…”.

c. Comment 2.3: Table 1 formatting

Reviewer's comment: “Table 1(General Characteristics): In the provided draft, this table's formatting is disrupted. Please ensure the final version presents this information in a clear, columnar format that is easy for readers to scan. Key columns should include: Study, Design, Population (n, age, % hypertensive), Intervention/Exposure, Comparator, and Outcomes Assessed.”.

Response:

We have restructured Table 1 with the recommended column organization to facilitate quick reference for researchers. The new format presents information in a clear, scannable columnar layout.

Specific changes:

• Table 1, pages 15-19, according to Reviewer 2's recommendations.

d. Comment 2.4: I² statistics in forest plots

Reviewer's comment: “Figures 2-5 (Forest Plots): These are essential and well-presented. A minor enhancement would be to ensure the I² statistic for heterogeneity is included in the figure caption or within the plot itself for immediate reference”.

Response:

Thank you for this suggestion. We have now explicitly included heterogeneity information in all figure captions. Figures 3 and 5 report within-meta-analysis heterogeneity (Tau², Chi², and I²). Figures 2 and 4 present subgroup analyses, and therefore report the test for subgroup differences (Chi² and I²), reflecting between-subgroup heterogeneity. This information is now clearly stated in the figure captions for immediate reference.

Specific changes:

• Figure 2 caption, page 11, line 342: Added: Test for subgroup differences: Chi2=5.66, df=1 (P=0.02), I2=82.3%

• Figure 3 caption, page 12, line 361: Added: Heterogeneity: Tau2= 0.00; Chi2=0.00, df=1 (P=0.98); I2=0%

• Figure 4 caption, page 20, line 512: Added: Test for subgroup differences: Chi2=4.92, df=1 (P=0.03), I2=79.7%

• Figure 5 caption, page 21, line 535: Added: Heterogeneity: Tau2= 0.00; Chi2=0.35, df=1 (P=0.55); I2=0%

3. Additional Editor Comments:

We sincerely appreciate the editor’s guidance. Each of the points raised has been considered and addressed in our responses to the reviewers, summarized below:

e. Comment 3.1. Clarification of control group composition

Editor's comment: “As the major revision is marked by one of the reviewers on the point that either the control is women using non hormonal contraceptive or women using no contraceptive. It should be clear and justified”.

Response:

We thank the editor for highlighting this. This issue was addressed in our response to Reviewer #1, Comment 1.1, where we clarified the composition of control groups and the limitations of the studies regarding safety and effectiveness outcomes.

f. Comment 3.2. Vote count methodology specification

Editor's comment: “The second review comments that there need to specify the specific rule for the vote count (e.g., direction of effect based on point estimate, statistical significance)”.

Response:

Thank you for this suggestion. This was addressed in our response to Reviewer #2, Comment 2.1, with explicit description of our vote counting rules in the Methods section.

g. Comment 3.3. Stylistic variation for certainty language

Editor's comment: “The statement by the authors that the evidence is uncertain should also replace with suitable synonyms”.

Response:

We appreciate this comment. This was addressed in our response to Reviewer #2, Comment 2.2, where we replaced repetitive uses of “the evidence is very uncertain” with suitable synonyms throughout the manuscript.

h. Comment 3.4. Table 1 formatting

Editor's comment: “Table 1 formatting needs correction so that it could be easily scanned for other researcher as reference. Key columns should include: Study, Design, Population (n, age, % hypertensive), Intervention/Exposure, Comparator, and Outcomes Assessed”.

Response:

Thank you for pointing this out. This was addressed in our response to Reviewer #2, Comment 2.3, and Table 1 has been reformatted for clarity and ease of reference.

i. Comment 3.5. I² statistics in forest plots

Editor's comment: “Figures 2-5: I² statistic for heterogeneity should included in the figure caption or within the plot itself for immediate reference”.

Response:

We thank the editor for this suggestion. This was addressed in our response to Reviewer #2, Comment 2.4, and all figure captions now include heterogeneity statistics (I², Tau², Chi²) as requested.

4. Final Remarks:

We sincerely appreciate the thoughtful feedback provided by the Editor and Reviewers. We believe that all concerns have been fully addressed and hope that the revised manuscript now meets the standards for publication in PLOS ONE.

Thank you very much for your time and consideration.

Sincerely,

Natalia Losada-Trujillo

on behalf of all authors

---

## [Editor Report · Decision Letter 2]

12 Mar 2026

Safety and effectiveness of hormonal vs non-hormonal or no contraception in women with hypertension and future fertility desire: a broad-scope systematic review

PONE-D-25-37651R2

Dear Dr. LOSADA TRUJILLO,

We’re pleased to inform you that your manuscript has been judged scientifically suitable for publication and will be formally accepted for publication once it meets all outstanding technical requirements.

Kind regards,

Noor Jahan

Academic Editor

PLOS One
---

## [Editor Report · Acceptance letter]

PONE-D-25-37651R2

PLOS One

Dear Dr. Losada-Trujillo,

I'm pleased to inform you that your manuscript has been deemed suitable for publication in PLOS One. Congratulations! Your manuscript is now being handed over to our production team.

Kind regards,

on behalf of

Prof. Dr. Noor Jahan

Academic Editor

PLOS One